# A novel implantable mechanism-based tendon transfer surgery for adult acquired flatfoot deformity: Evaluating feasibility in biomechanical simulation

**Hantao Ling** *, **Ravi Balasubramanian**

School of Mechanical, Industrial, and Manufacturing Engineering, Oregon State University, Corvallis, Oregon, United States of America

* lingh@oregonstate.edu

**Data Availability Statement:** All OpenSim models, setup files, muscle activation files, data, and MATLAB scripts used to process the data are available from the Dryad data repository https://

## Abstract

Adult acquired flatfoot deformity becomes permanent with stage III posterior tibialis tendon dysfunction and results in foot pain and difficulty walking and balancing. To prevent progression to stage III posterior tibialis tendon dysfunction when conservative treatment fails, a flexor digitorum longus to posterior tibialis tendon transfer is often conducted. However, since the flexor digitorum longus only has one-third the force-capability of the posterior tibialis, an osteotomy is typically also required. We propose the use of a novel implantable mechanism to replace the direct attachment of the tendon transfer with a sliding pulley to amplify the force transferred from the donor flexor digitorum longus to the foot arch. In this work, we created four OpenSim models of an arched foot, a flatfoot, a flatfoot with traditional tendon transfer, and a flatfoot with implant-modified tendon transfer. Paired with these models, we developed a forward dynamic simulation of the stance phase of gait that reproduces the medial/lateral distribution of vertical ground reaction forces. The simulation couples the use of a fixed tibia, moving ground plane methodology with simultaneous activation of nine extrinsic lower limb muscles. The arched foot and flatfoot models produced vertical ground reaction forces with the characteristic double-peak profile of gait, and the medial/lateral distribution of these forces compared well with the literature. The flatfoot model with implant-modified tendon transfer produced a 94.2% restoration of the medial/lateral distribution of vertical ground reaction forces generated by our arched foot model, which also represents a 2.1X improvement upon our tendon transfer model. This result demonstrates the feasibility of a pulley-like implant to improve functional outcomes for surgical treatment of adult acquired flatfoot deformity with ideal biomechanics in simulation. The real-world efficacy and feasibility of such a device will require further exploration of factors such as surgical variability, soft tissue interactions and healing response.

datadryad.org/stash/share/oUWcB5OFab_
DPK183T6rwoTEAE1vZfGJfZNsU9y-y3g.

**Funding:** The author(s) received no specific funding for this work.

**Competing interests:** The authors have declared that no competing interests exist.

## Introduction

Traumatic injury or degeneration of the posterior tibialis (PT) tendon causes the foot arch to collapse and is one of the primary causes of adult acquired flatfoot deformity [1]. The PT tendon runs past the medial malleolus, inserts into the navicular tuberosity on the medial side of the foot, and acts as a key supporter of the medial longitudinal arch (see Fig 1A) [1, 2]. In stage III PT tendon dysfunction, as classified by Johnson and Strom [3], the flatfoot condition becomes permanent. This leads to foot pain, difficulty walking and balancing, and anatomical deformity. To avoid progression to stage III, conservative treatment, such as footwear and orthotics, are prescribed in stage II PT tendon dysfunction. However, in instances where conservative treatment is ineffective, surgical intervention is indicated.

One of the common surgical procedures employed in the treatment of stage II PT dysfunction is a flexor digitorum longus (FDL) to PT tendon transfer surgery followed by a corrective osteotomy [1, 4–7]. In the tendon transfer surgery, the FDL tendon is separated from its natural insertion point on the lateral toes and directly attached to the navicular tuberosity to re-establish supporting forces to the medial longitudinal arch (see Fig 1B). Due to this direct coupling between the tendon and bone, the force transferred from the donor muscle to the host site is limited by the donor muscle's force generating capacity; in this case, the FDL. Since the FDL generates less than one-third the force of the PT, the tendon transfer only partially restores the physiological supporting forces to the foot arch. Typically, the lack of strength of the transferred FDL is accepted because its primary function is to oppose foot eversion generated by the peroneus brevis muscle, which the FDL has two-thirds the strength of [8]. However, while the tendon transfer procedure alone leads to pain relief and improved inversion strength, the deformity often remains. This suggests that additional strength may be required in the transferred FDL to properly restore the foot arch and correct adult acquired flatfoot deformity. Due to the lack of candidate donor muscles and limitations in soft tissue surgery options, an additional osteotomy is currently required to fully correct the deformity and to optimize the biomechanics of the reconstructed PT tendon [1].

To address the limitations that arise in orthopedic surgery from using direct attachments, such as sutures, bone screws, and suture anchors, our group has previously developed a separate rod-shaped implantable mechanism that improves grasp function for patients with high median-ulnar nerve palsy [9, 10]. The device is implanted in the forearm between hand flexor tendons and redistributes the movement generated by a single muscle to multiple tendons by creating a "differential" tendon network.

In this paper, we explore the feasibility of a different type of passive implantable mechanism in the early developmental stage that amplifies muscle force for the treatment of adult acquired flatfoot deformity. The novel implant will amplify the force transferred from the weaker FDL to the PT tendon by replacing the direct attachment between the tendons with a pulley-like

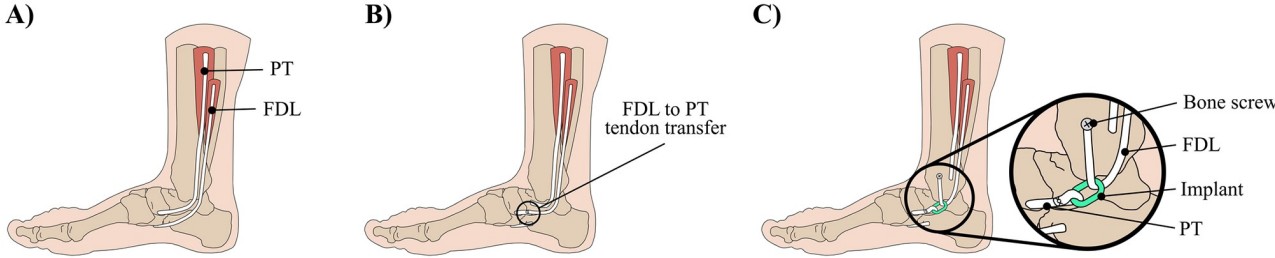

**Fig 1. Treatment options for adult acquired flatfoot deformity.** Drawings of (A) an arched foot, (B) a traditional FDL to PT tendon transfer surgery, and (C) our proposed implant-modified FDL to PT tendon transfer surgery.

device that provides mechanical advantage (see Fig 1C). Since the implant is passive, it eliminates the need for external power, electronics, motors, and control signals. Thus, the implant will enable the individual's own muscle to generate the force and movement necessary to overcome the strength difference between the FDL and PT. If this device can be successfully developed and demonstrate efficacy in foot arch restoration, the force amplification provided by the implant has potential to restore supporting forces to the medial longitudinal arch beyond what is attainable with current surgical techniques. For the appropriate patients, the need for an additional osteotomy may be completely supplanted with the implant-based tendon transfer procedure.

A preliminary human cadaver study conducted using this concept yielded promising results, with the implant-modified procedure leading to a significant medial shift of the center of pressure during the stance phase of gait [11]. However, evaluating the efficacy of surgical procedure and optimizing the design of surgical implants in cadaveric specimens is an extremely tedious process. Biomechanical simulations enable rapid design modification and provide immediate feedback regarding parameter sensitivity and post-operative outcomes. Since additional aspects of the implant design and surgical protocol need to be explored to fully evaluate the biomechanical feasibility of this implant-based procedure, this motivates the development of a biomechanical simulation capable of incorporating varying surgical protocols and foot pathologies while providing a measure of functional patient outcomes. We are especially interested in evaluating vertical ground reaction force distributions, which are a key parameter in both diagnosing foot pathologies [12–15] and retrospectively evaluating treatment efficacy [16, 17].

Surgical treatment of stage II adult acquired flatfoot deformity results in a quantifiable lateral shift of vertical ground reaction forces during the stance phase of gait [11, 18, 19]. However, there is currently a lack of lower extremity models that are capable of producing ground reaction force distributions during the stance phase of gait as outputs to simulation. Instead, ground reaction forces are primarily recorded experimentally and used as inputs in biomechanical simulations to estimate muscle forces and joint kinematics in inverse dynamic simulations [20, 21]. Thus, the models produced in this work seek to accurately generate vertical ground reaction forces of gait as an output variable to enable us to predict treatment outcomes under ideal circumstances defined in forward dynamic biomechanical simulation.

Furthermore, a review of forty-one manuscripts showed that the current models did not emphasize the foot arch in dynamic gait simulation [22]. Specifically, current foot models have either three or four segments that include the hindfoot, midfoot, forefoot, and hallux, which only divide the foot into anterior and posterior sections. Without a degree of freedom between the medial and lateral sides of the foot, the height of the medial longitudinal arch cannot be easily adjusted between models, and the deformations of the arch cannot be properly accounted for in gait simulation. While models with a fully-defined medial longitudinal arch and passive ligaments do exist, these models are typically loaded statically and are not simulated in gait due to their complexity [23–25]. Modeling all the soft tissue and rigid body interactions in these complex models lead to heavy computational and time resource usage if applied to dynamic gait simulation. Conversely, simple models that are capable of accurately reproducing ground reaction forces during gait typically lack the detail to enable complex model modification to analyze various surgical treatment options or foot pathologies [26–30]. With reproduction of ground reactions forces as a main target for these models, anatomical definition is often abstracted away in favor of simplified mathematical models that reduce the computational strain and complexity in simulating dynamic gait.

The aim of this paper is to create a forward dynamic biomechanical simulation of gait to determine the feasibility and efficacy of using a force-amplifying implantable mechanism in

restoring the foot arch for individuals with adult acquired flatfoot deformity. The biomechanical models presented in this paper differ from previously developed gait models because they use forward dynamics to produce ground reaction forces throughout the stance phase of gait rather than use ground reaction force data as an input to inverse dynamic simulation. Our multi-segmented models also use a simplified medial longitudinal arch to enable arch height adjustment between models and arch deformation during gait simulation. This model feature is especially important since the severity of adult acquired flatfoot deformity correlates with a decrease in medial longitudinal arch height, which alters the medial/lateral distribution of vertical ground reaction forces over the plantar aspect of the foot throughout the stance phase of gait.

In total, we have developed four foot models for simulation: (1) an arched foot, (2) a flatfoot with stage II adult acquired flatfoot deformity (henceforth called "flatfoot"), (3) a flatfoot with the traditional FDL to PT tendon transfer, and (4) a flatfoot with the implant-modified tendon transfer. These models were developed by modifying an existing lower limb model in Open-Sim 3.3. OpenSim is a biomechanical simulation platform that has been used extensively to model lower extremity kinetics and kinematics [31–34]. The fidelity of the ground reaction force data generated by our simulations and models were compared with experimental results from prior work in the field. Specifically, we compare the overall ground reaction forces generated by the arched foot and flatfoot models to the literature, and we evaluate the differences in the medial/lateral distribution of vertical ground reaction forces created by our four foot models. The biomechanical efficacy of using a novel force-amplifying implant to treat adult-acquired flatfoot deformity was analyzed by comparing ground reaction force data generated by our implant-modified tendon transfer flatfoot model with data from the other three foot models.

## Materials and methods

The lower-extremity models presented in this paper are based on cadaveric models from a collaborative study using a "Robotic Gait Simulator" to evaluate the feasibility and efficacy of a pulley-like device similar to the implant described in this paper [11]. The Robotic Gait Simulator is a six degree-of-freedom robot that reproduces the kinematics and kinetics of the stance phase of human gait in lower limb cadaver specimens [35]. The robot has been used previously to simulate both physiological and pathological gait [36, 37]. It replicates gait by actuating a ground plane with the inverse motion of gait relative to a lower-limb cadaveric specimen fixed at the tibia. The ground plane is instrumented with a force plate that records ground reaction forces through the simulated stance phase of gait. While the gait kinematics are reproduced through the ground plane, linear actuators simultaneously coordinate the "activation" of nine lower limb tendons: the PT, FDL, anterior tibialis, extensor digitorum longus, extensor hallicus longus, peroneus longus, peroneus brevis, flexor hallicus longus, and the Achilles. In the cadaver study, stage II adult acquired flatfoot deformity models were simulated by collapsing the medial longitudinal arch through ligament attenuation followed by cyclic weight-bearing loading. Three of the experimental groups developed from this cadaveric flatfoot model were an untreated flatfoot, a flatfoot treated with a FDL to PT tendon transfer surgery, and a flatfoot treated with an implant-modified tendon transfer surgery.

The computational models described in this paper seek to replicate these three flatfoot models and provide an additional arched foot model for comparison. Like the Robotic Gait Simulator, the biomechanical simulation discussed in this paper uses a moving ground plane, lower limb models fixed at the tibia, and simultaneous activation of nine extrinsic lower limb muscles while recording ground reaction forces between the foot models and the ground

plane. These modifications greatly simplify a full-body gait model down to a single limb actuated by nine muscles while retaining the complex kinematics and kinetics associated with ground reaction force generation during the stance phase of gait (see Fig 2). The two primary components of the biomechanical simulation are (1) the foot models and (2) the simulation of

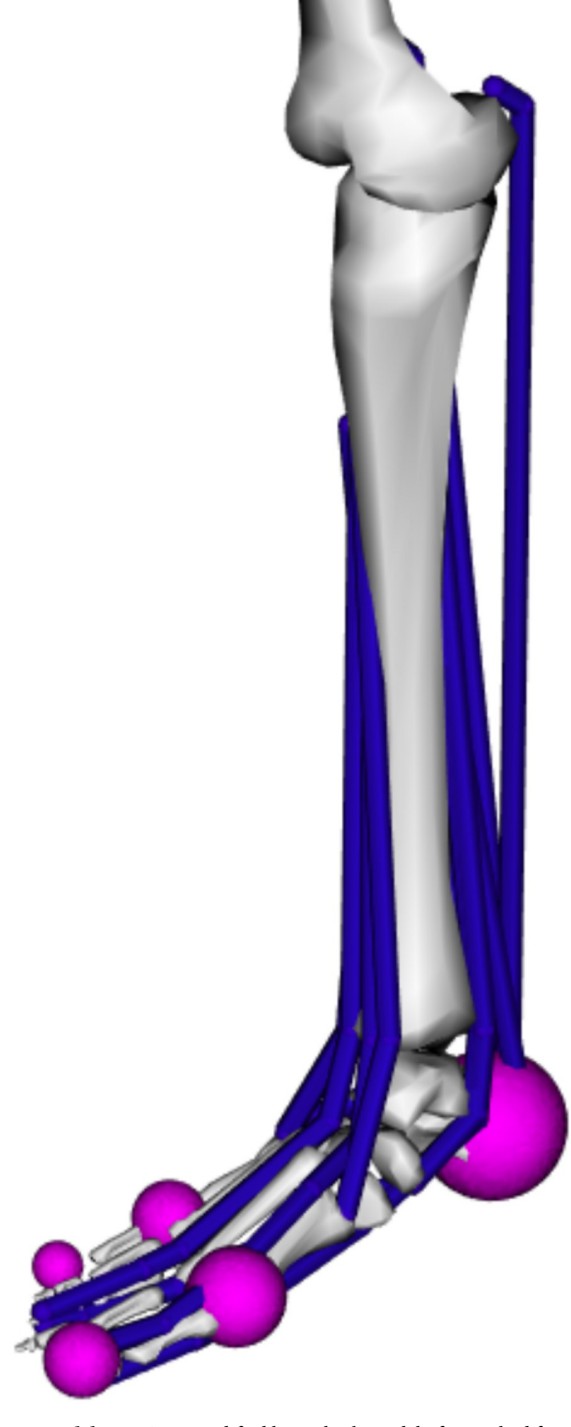

**Fig 2. Foot model overview.** Modified lower limb model of an arched foot truncated at the femur with nine extrinsic muscles (blue) and five spheres (pink) for generating contact with a moving ground plane.

gait kinematics and kinetics. Further details on the development of these components are described in the following sections.

## Foot models

The four foot models presented in this paper (arched foot, flatfoot, flatfoot with tendon transfer surgery, and flatfoot with implant-modified tendon transfer surgery) are modified versions of Stanford's Gait2392 full-body model. The original OpenSim model represents a healthy adult subject with normal joint function and without adult acquired flatfoot deformity, so the foot models presented in this work are constrained by the base model they are developed from. Since the properties of the original model were not designed to replicate pathological function, the expected compensatory adaptions of tendons and muscles to adult acquired flatfoot deformity are not represented. However, this limitation does not prevent between-model comparisons from being drawn from the simulations.

The Gait2392 model contains seventy-six muscles in the torso and lower extremities [31]. Inertial properties of the bones were inherited from the original model, which uses adapted experimental values from a previously developed computational model [38]. To minimize simulation complexity, our models are truncated at the knee joint and include bones from only the right lower limb. We can still fully simulate the stance phase of gait with this simplification because our methodology takes advantage of foot models fixed in space at the tibia while a moving ground plane tracks the inverse motion of gait. This modification reduces the seventy-six muscles in the original model to nine extrinsic muscles of the lower limb: the PT, FDL, anterior tibialis, extensor digitorum longus, extensor hallicus longus, flexor hallicus longus, peroneus longus, peroneus brevis, and soleus.

Four joints are included in each of the foot models. The ankle, subtalar, and metatarsophalangeal joints are inherited from the original Gait2392 model while a simplified medial longitudinal arch joint was created as part of this work. The three joints included with the original model rotate about axes defined in the original model [31]. The foot arch joint is parallel to the sagittal plane in the midfoot and divides the first metatarsal and phalanx from the second to fifth metatarsals and phalanges. This additional arch joint combined with the model's existing metatarsophalangeal joint results in four distinct sections of the foot (see Fig 3). The medial

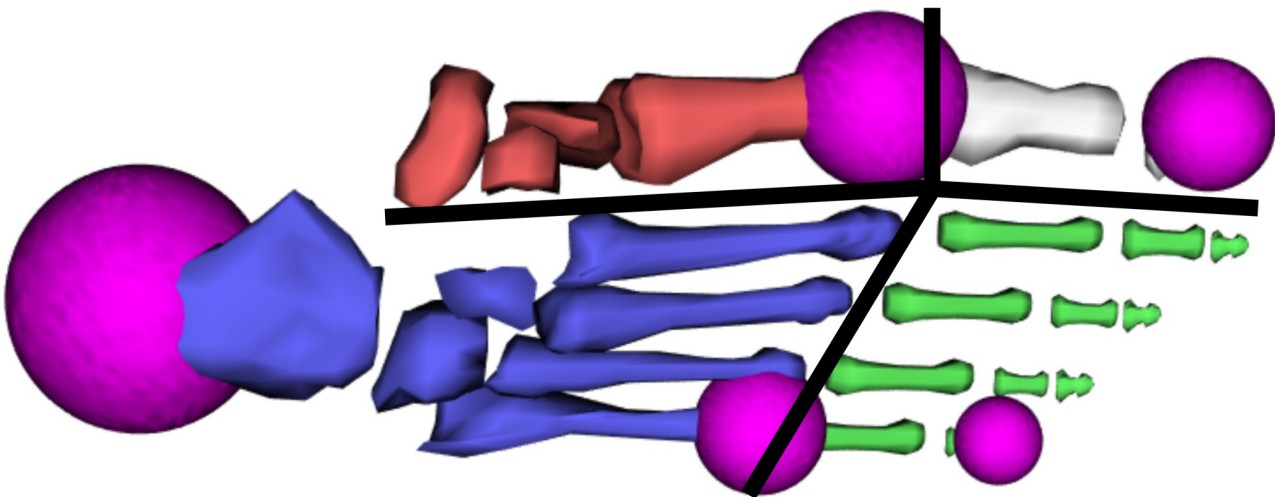

**Fig 3. Foot model segments and contact geometries.** Foot model developed for the gait simulations with four distinct segments: medial midfoot (red), medial forefoot (white), lateral midfoot (blue), and lateral forefoot (green). Five spheres used for generating contact forces between the foot and ground plane (pink) and the joints between foot segments (black lines) are also shown.

(first phalanx to navicular) and lateral (second through fifth phalanges to calcaneus) halves of the foot can rotate independently from each other about the foot arch joint. The first phalanx and second through fifth phalanges can flex independently of each other about the metatarso-phalangeal joint, which was split at the intersection of the arch joint. Decreasing the angle of the arch joint raises the medial longitudinal arch while increasing the angle relaxes the medial longitudinal arch and flattens the foot.

In the arched foot model, the arch joint angle was maintained from the original Gait2392 model. In the flatfoot models, the arch joint was flattened according to an 8 mm difference in first metatarsocuneiform height typically observed between arched feet and flatfeet [6, 39]. In addition to considering the skeletal geometry in our simplified foot arch model, we also included a simple model of the medial longitudinal arch's elasticity, which is crucial for energy storage during gait [40]. The foot arch's elasticity is typically provided by the passive deltoid and spring ligaments in the foot and the attachment of the posterior tibialis muscle onto the navicular bone [41]. In our foot models, we simplify the elasticity produced by the passive foot ligaments with a torsional stiffness of 1 Nm/˚ applied at the arch joint.

For the four foot models, the paths of the nine extrinsic muscles remained consistent with the original Gait2392 model. Modifications to the PT and FDL muscles were made when appropriate to physiologically replicate each condition. In the arched foot model, no modifications were made to the routing of the muscles and tendons (see Fig 4A). In the flatfoot model, the PT muscle was disabled to simulate PT tendon dysfunction in stage II adult acquired flatfoot deformity, resulting in no PT muscle force being transferred to the navicular bone (see Fig 4B). In the flatfoot model with tendon transfer surgery, the FDL muscle was disabled to simulate removal of the FDL tendon from its original insertion. FDL forces were instead applied through the PT muscle, which maintained its original muscle path terminating on the navicular bone. Consequently, the original PT now acts as the transferred FDL in the tendon transfer model (see Fig 4C).

In the flatfoot model with implant-modified tendon transfer surgery, the FDL muscle was again disabled to simulate its use as the donor tendon in tendon transfer surgery. The PT, which now acts the transferred FDL, was rerouted to represent the implant-modified tendon transfer. Rather than route the transferred FDL directly to the navicular bone, it is routed around the pulley implant, pulled back parallel with the proximal portion of the transferred FDL, and then anchored onto the tibia. The original tendon length of the transferred FDL was maintained during modification of the muscle path. An artificial tendon was then introduced into the model with one end fixed on the implant and the other end attached to the original insertion point of the PT tendon on the navicular bone (see Fig 4D). Due to the degradation of

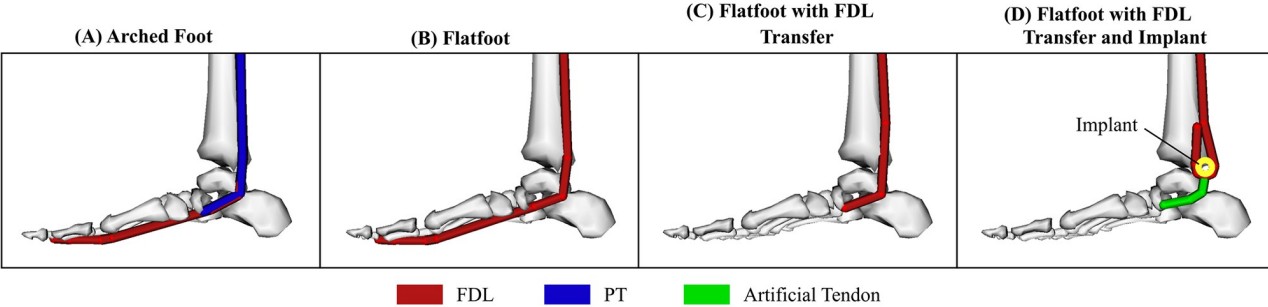

**Fig 4. Overview of the four foot models.** The stance phase of gait was simulated with forward dynamics using (A) an arched foot model, (B) a flatfoot model, (C) a flatfoot with FDL tendon transfer model, and (D) a flatfoot with FDL tendon transfer and implant model. Only the routing of the flexor digitorum longus (FDL) and posterior tibialis (PT) are shown for clarity.

the original PT tendon, this artificial tendon in the foot models represents either grafted tendon or artificial tendon that will be required to make this attachment clinically. The movement of the implant within this model is constrained to the sagittal plane. This simplification was made because we assume the implant operates in its intended design without twisting or rotating and to reduce simulation complexity. After the transfer, FDL activation will cause its tendon to pull and slide on the implant, resulting in proximal translation of the implant. This artificially-created tendon network now acts like a pulley system and amplifies the force generated by the FDL, which is then transferred to the foot arch (see Fig 5). Fundamental physics dictates that the trade-off for the mechanical advantage provided by the implant is a proportional loss of excursion. Thus, 2X the excursion of the transferred FDL tendon is required to produce 2X the force at the insertion of the PT tendon and the same excursion as the original PT. In the ideal conditions created by this model, the implant creates a force amplification of 2X. In reality, the force amplification depends on several parameters, including FDL anchoring location, anchoring method, and tendon-implant friction. However, these parameters are beyond the scope of this paper, which aims to purely evaluate the initial feasibility and efficacy of such an implant under ideal circumstances in biomechanical simulation.

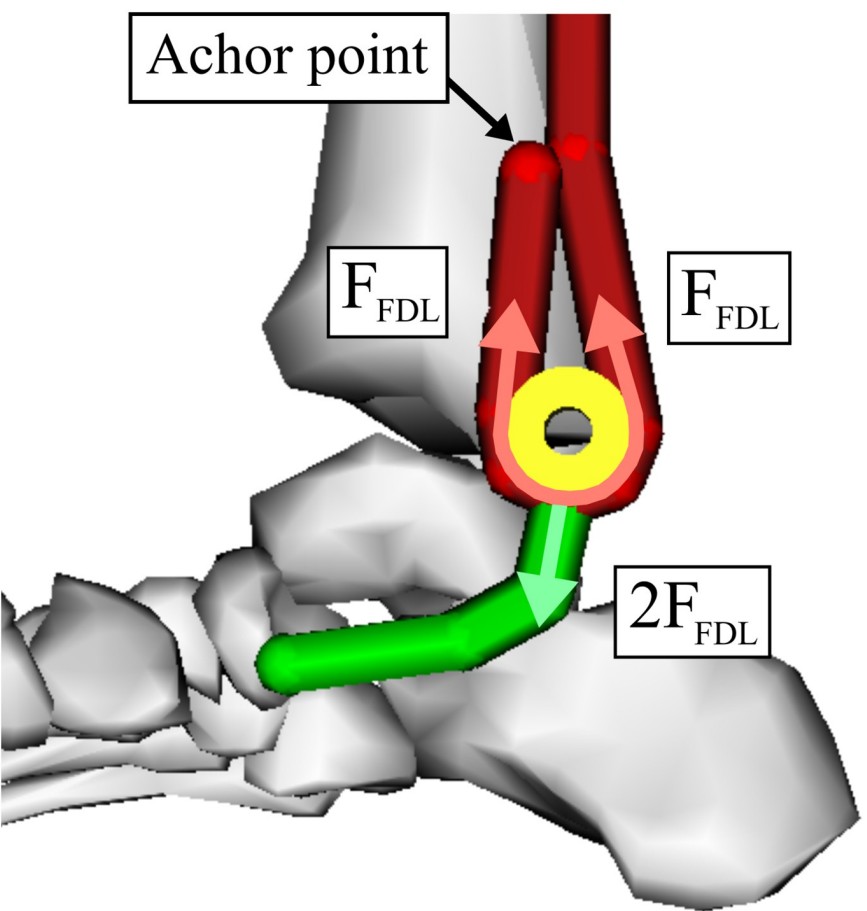

**Fig 5. Diagram of the force amplification created by the implant-based tendon transfer surgery.** The tendon network created by the implant forms a pulley system that amplifies the force generated by the transferred flexor digitorum longus muscle as it is routed to the navicular bone.

The model parameters that were not mentioned as modifications of the original Gait2392 model, such as bone geometry and muscle force-length curves, were kept consistent across all four models described in this paper.

## Gait simulation kinetics and kinematics

To generate vertical ground reaction force distributions for each of the four foot models, a simulation of the stance phase of gait was developed. The simulation consists of three primary components: coordinated muscle activations, ground plane kinematics, and contact dynamics between the ground plane and foot.

During the simulated stance phase of gait, muscle forces are generated for the nine extrinsic muscles in our foot models (PT, FDL, anterior tibialis, extensor digitorum longus, extensor hallucis longus, flexor hallucis longus, peroneus longus, peroneus brevis, and soleus). The muscle forces are derived from electromyography-based estimates of muscle activation from a healthy population defined in previous studies using the Robotic Gait Simulator [35, 37, 42, 43]. The activation levels were defined at ten evenly distributed points throughout the stance phase of gait. The time histories of the muscle forces for the arched foot were generated in OpenSim for the entire stance phase of gait by linearly interpolating between these ten points and taking a product between each muscle's activation curves and their maximum isometric force (see Fig 6). While the piecewise linear nature of these presented muscle forces do not mimic physiological muscle force patterns, we prefer to use this low resolution data because it is coupled with the kinematic data from the Robotic Gait Simulator studies we are mirroring with our simulations. We recognize that this is a limitation of our study. However, because all

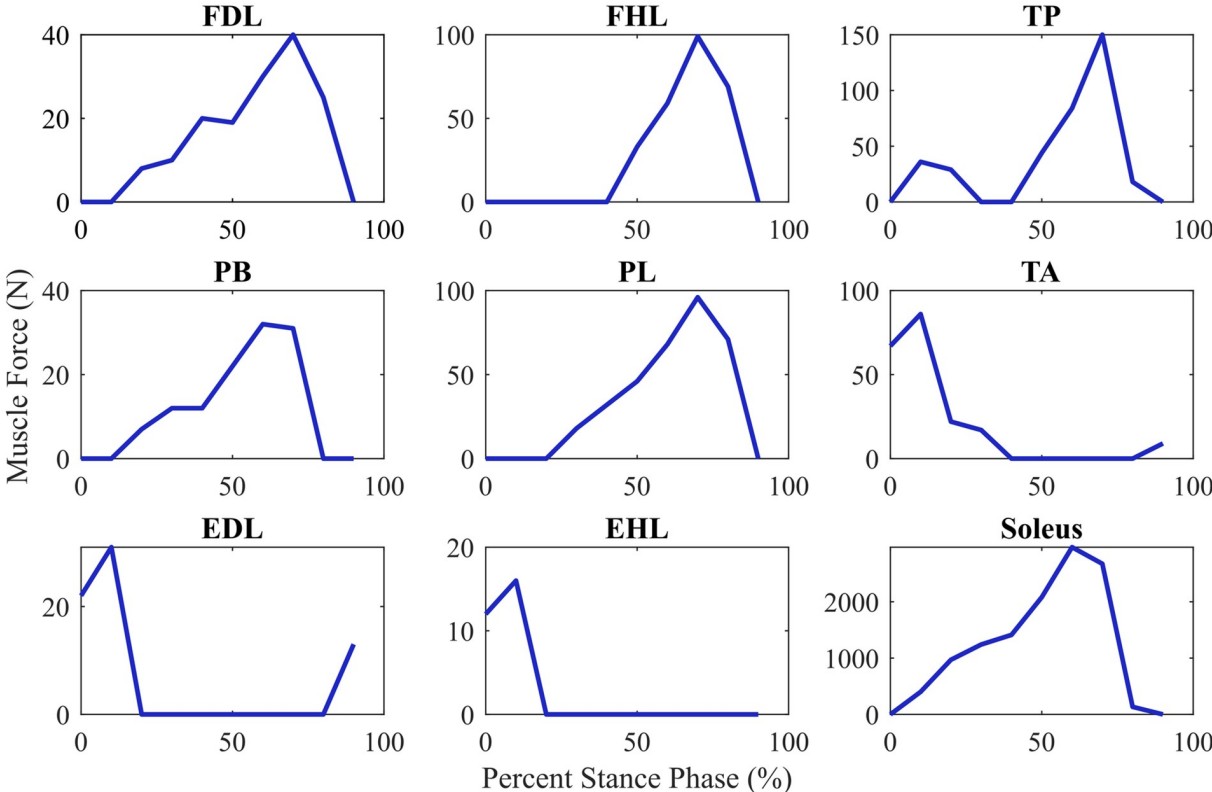

**Fig 6. Generated muscle forces.** Forces generated by nine extrinsic muscles of the lower limb during simulation of the stance phase of gait.

four of our foot models are actuated with the same methodology, we remain confident in our ability to make between-model comparisons.

In the flatfoot model, the same muscle activations as the arched foot were used with the exception of the PT, which was not activated to simulate PT tendon dysfunction [11, 44]. While compensatory muscle activations are likely to develop in adult acquired flatfoot deformity, directly removing the PT forces in this manner enables us to isolate the effects that a dysfunctional PT tendon may have on shifting vertical ground reaction forces during gait compared to our arched foot model.

In the two treatment models (flatfoot with tendon transfer and flatfoot with implant-modified tendon transfer), the FDL muscle forces shown in Fig 6 were applied to the original PT muscle path to represent the new transferred FDL tendon. In this scenario, we assume that there are no negative effects of tendon re-routing introduced during surgery for the flatfoot with traditional tendon transfer surgery model because the path of the transferred FDL exactly matches that of the original PT. This enables us to evaluate the efficacy of our implantable device compared to an ideal surgical scenario. The original FDL muscle was not activated, and all other muscles were actuated with the same activation curves as previously discussed for the arched foot and flatfoot models. By maintaining these muscle activation curves across the four models, we can more clearly evaluate how the FDL tendon transfer and implant-modified tendon transfer models shift vertical ground reaction force distributions during gait when compared to our base flatfoot and arched foot models.

These simplifications in muscle activations and tendon routing in our models enable us to initially determine the feasibility and efficacy of a force-amplifying implant when compared to similar models under ideal circumstances.

Since all the foot models are fixed in space at the tibia, gait kinematics are reproduced by applying the inverse motion of the stance phase of gait through a rectangular ground plane. This decision to generate gait kinematics through the ground plane greatly simplifies our simulation because we circumvent the need to control the complex dynamics of accelerating a full-body skeletal model. Instead, the simple ground plane drives gait kinematics, so we can reduce the full-body model to a single lower-limb skeletal model. Because of this methodology, full dorsiflexion to plantarflexion range of motion is achieved in simulation even with the excursion trade-off produced by the implant mechanics. While the active range of motion of the foot models is not explored in this study, a separate gait study using the Robotic Gait Simulator and human lower limb cadavers showed that the implant-based procedure did not cause the FDL muscle to exceed its physiological excursion limits [11].

The prescribed motion of the ground plane was generated by fitting polynomial curves to experimental data tracking three translational and three rotational coordinates from a previous cadaver study of adult acquired flatfoot deformity using the Robotic Gait Simulator [11, 35]. The ground plane began in contact with the heel (heel-strike) and ended in contact with the hallux (toe-off). The simulated stance phase kinematics are based off the cadaver study and lasts 4.09 s. This duration is about one-sixth the speed of physiologic gait and was originally used to protect the integrity of the cadaveric specimens [11]. The same kinematic trajectory of the ground plane was used in the simulation of all four foot models to remain consistent with the experimental methods used by the Robotic Gait Simulator and is shown in Fig 7. A visual progression of the arched foot model through the described kinematic trajectory of the ground plane is provided in Fig 8.

The contact between the ground plane and the foot models was simulated using an elastic foundation force contact model [45]. This model uses a layer of triangular meshes covered with springs that independently interact with other objects in contact with the mesh, which enables application of both stiffness and friction characteristics. Stiffness values for the contact

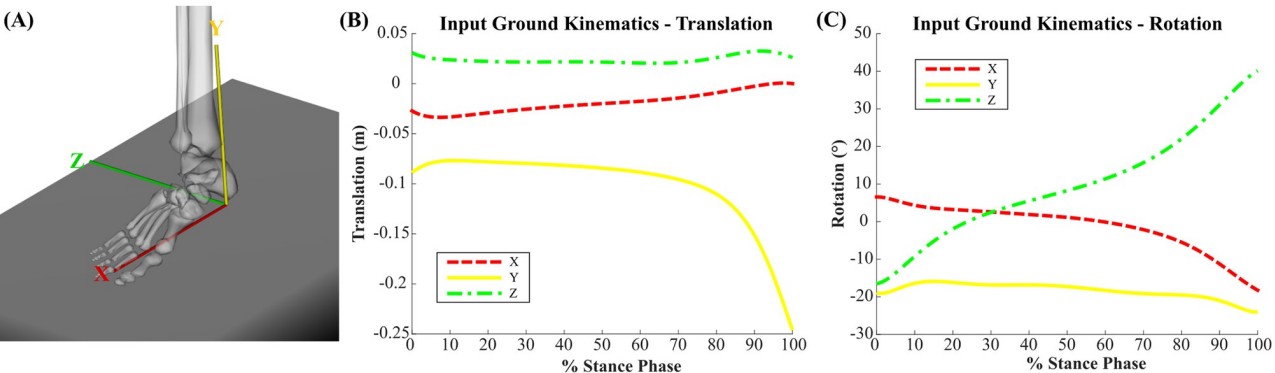

**Fig 7. Ground plane kinematics.** Kinematic trajectory of the ground plane used as an input to gait simulation. (A) shows the local coordinate axes of the ground plane with respect to the orientation of a foot model. The input kinematic trajectory is split into (B) translational and (C) rotational components.

objects were adjusted until realistic deformations of the foot pads during gait were attained (peak of approximately 1 cm [46–48]). We assumed that the foot would not slip while in contact with the ground, so we used a very high friction coefficient to prevent slip between contact spheres on the foot models and the ground plane and to simplify computation during simulation. A viscous friction model is provided in OpenSim and is defined as $F_F = F_N F_v v_s$, where $F_N$ is normal force, $F_v$ is the coefficient of viscous friction, and $v_s$ is the slip velocity.

Contact forces were defined between designated contact objects placed on both the ground plane and the foot models. For the ground plane, a rectangular contact object with the same dimensions as the ground plane was defined as the contact object. For each foot model, five contact spheres were placed on the foot's plantar aspect (see Fig 3). The sizing and placement of the contact spheres were approximated based on ellipsoidal contact volumes on the foot model developed by Brown and McPhee that reproduces ground-contact forces in forward dynamic simulation [48]. In their model, the size of the ellipsoids was determined based on a 70 kg subject. Since our model represents a larger, 82 kg subject, we increased the size of each of our spheres by 1 cm. We also split the metatarsal ellipsoid created in their work into two individual spheres and added an additional sphere at the fifth toe to enable evaluation of the medial/lateral distribution of vertical ground reaction forces. Specifically, we placed one 6 cm sphere at the heel while the four remaining spheres were placed to measure contact in each of

| Heel Strike | Early Midstance | Late Midstance | Toe-Off |
|---|---|---|---|

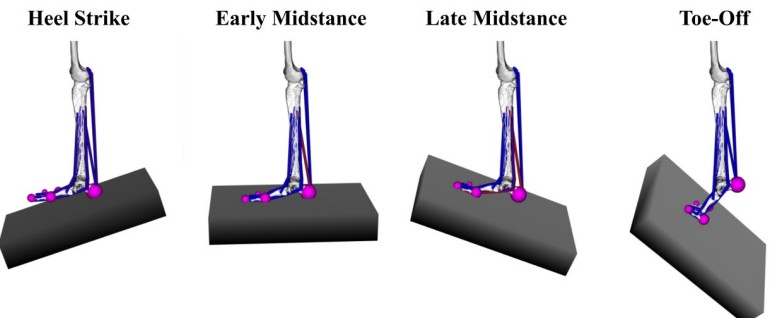

**Fig 8. Simulation of stance phase of gait.** Gait simulation kinematics of the stance phase of gait through a healthy foot model. The tibia of the model is held fixed while a moving ground plane tracks the inverse motion of gait at heel strike, early midstance, late midstance, and toe off.

the four foot segments at the first metatarsal head (4 cm), fifth metatarsal head (3 cm), hallux (3 cm), and fifth toe (2 cm). The placement of the spheres creates contact points around the foot perimeter and between each of the four segments of the foot model and the ground plane.

A body weight loading of approximately 82 kg was applied to each of the foot models. This was accomplished by iteratively adjusting the initial distance between the foot models and the ground plane. At zero-percent stance phase, the heel contact sphere was set to intersect the ground plane such that the magnitude of the double-peak profile of vertical ground reaction forces represented that of an 82 kg individual's gait. Contact between the foot models and the ground plane was maintained throughout stance phase via the ground plane kinematic trajectory, muscle activations, and friction between foot models and the ground plane. No other soft tissue interactions were included as part of the simulation. This contact model for the foot greatly simplifies the complex contact dynamics of gait while enabling the focus of this paper; namely, the evaluation of the medial-lateral distribution of vertical ground reaction forces throughout the stance phase of gait [49].

## Results

The simulations presented in this paper produced two results: (Result 1) an analysis of our simulation methodology for generating vertical ground reaction forces during the stance phase of gait using our arched foot and flatfoot models; and (Result 2) a predictive comparison of treatment efficacy between a traditional tendon transfer procedure and our implant-based procedure for adult acquired flatfoot deformity. All ground reaction forces are normalized by body weight.

### Vertical ground reaction force generation analysis

The arched foot and flatfoot model simulations for generating vertical ground reaction forces were evaluated using two parameters: the overall vertical ground reaction force curves that were produced and the difference in the distribution of vertical ground reaction forces between the medial and lateral sides of the foot.

The overall vertical ground reaction forces generated by our simulations exhibit the characteristic double-peak profile of human gait (see Fig 9). They were plotted as the sum of vertical forces recorded by the five contact spheres on each foot model. In the arched foot simulation, the first peak in vertical ground reaction force occurs at 29.7% stance phase with a magnitude of 1.14 times body weight. In the flatfoot simulation, the first peak occurs at 30.8% stance phase with a magnitude of 1.09 times body weight. The second peak for the arched foot simulation appears with a magnitude of 1.07 times body weight at 73.0% stance phase, while the second peak in the flatfoot simulation appears with a magnitude of 1.16 times body weight at 72.3% stance phase. The minimum between the two peaks occurs at approximately 50% stance phase for both foot models at magnitudes of 0.900 times body weight for the arched foot and 0.905 times body weight for the flatfoot. These results are in close agreement with experimental studies from the literature [27, 49–55].

The distribution of vertical ground reaction forces between the medial and lateral sides of the arched foot and flatfoot models are displayed in Fig 10. These force curves are separated into pairs of measurements: medial and lateral vertical ground reaction forces for each foot model. The medial split of vertical ground reaction forces was plotted as a sum of the vertical forces generated by the contact spheres on the head of the first metatarsal and the hallux. The lateral split of vertical ground reaction forces was plotted as a sum of the vertical forces generated by the contact spheres on the head of the fifth metatarsal and the fifth toe. The forces

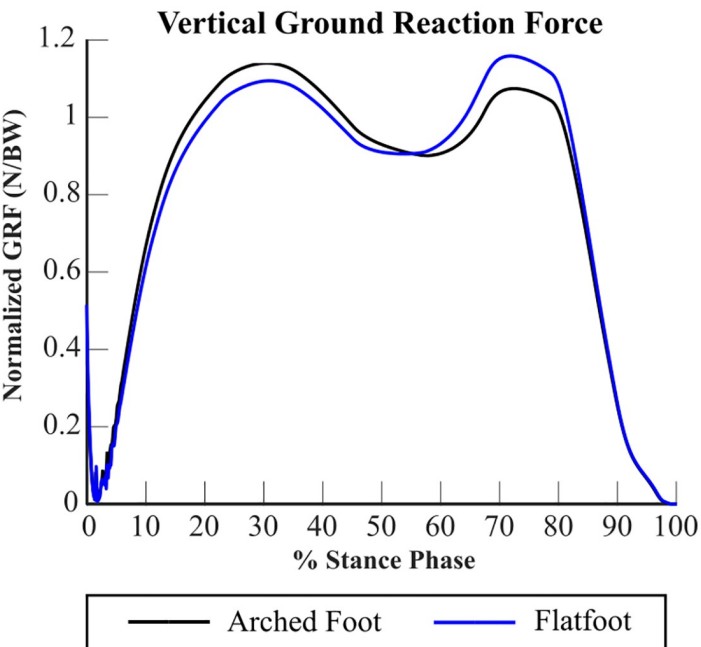

**Fig 9. Vertical ground reaction forces.** The vertical ground reaction forces generated by simulations of the stance phase of gait with an arched foot (black) and a flatfoot (blue) model.

generated by the contact sphere on the heel were identical across the models and matched the magnitude and timing seen in a previous study [56].

In the arched foot simulation, the medial and lateral vertical ground reaction forces peaked at 0.397 and 0.603 times body weight, respectively, which results in a 39.7%/60.3% medial/lateral split in peak vertical ground reaction forces. The flatfoot model generated peak medial and lateral vertical ground reaction forces of 0.493 and 0.507 times body weight, respectively, resulting in a 49.3%/50.7% medial/lateral split in peak vertical ground reaction forces. These medial/lateral splits in vertical ground reaction forces for the arched foot and flatfoot models were similar to those found in previous work [56–59]. Furthermore, these results indicate a 9.6% medial shift in peak vertical ground reaction forces between the arched foot model and the flatfoot model during the stance phase of gait, which agrees well with the 9.1% medial shift reported in an experimental study by Neville et al. with similar methodology [60].

## Treatment comparison

Fig 10 also displays the medial/lateral split of vertical ground reaction forces over the stance phase of gait for the flatfoot with tendon transfer surgery and the flatfoot with implant-modified tendon transfer surgery models. Medial and lateral forces for the two treatment models were calculated in the same manner as described for the arched foot and flatfoot models.

In the simulation using the flatfoot model with tendon transfer surgery, a 44.4%/55.6% medial/lateral split in peak vertical ground reaction forces was observed over the stance phase of gait. This distribution of vertical ground reaction forces indicates that the traditional tendon transfer surgery shifts foot plantar forces laterally by 3.9% when compared to no treatment (48.3%/51.7% medial/lateral split). The extent of restoration was calculated by dividing the lateral shift of ground reaction force found in the treatment model (3.9%) by the total medial shift of ground reaction force between the arched foot model and the flatfoot model (8.6%).

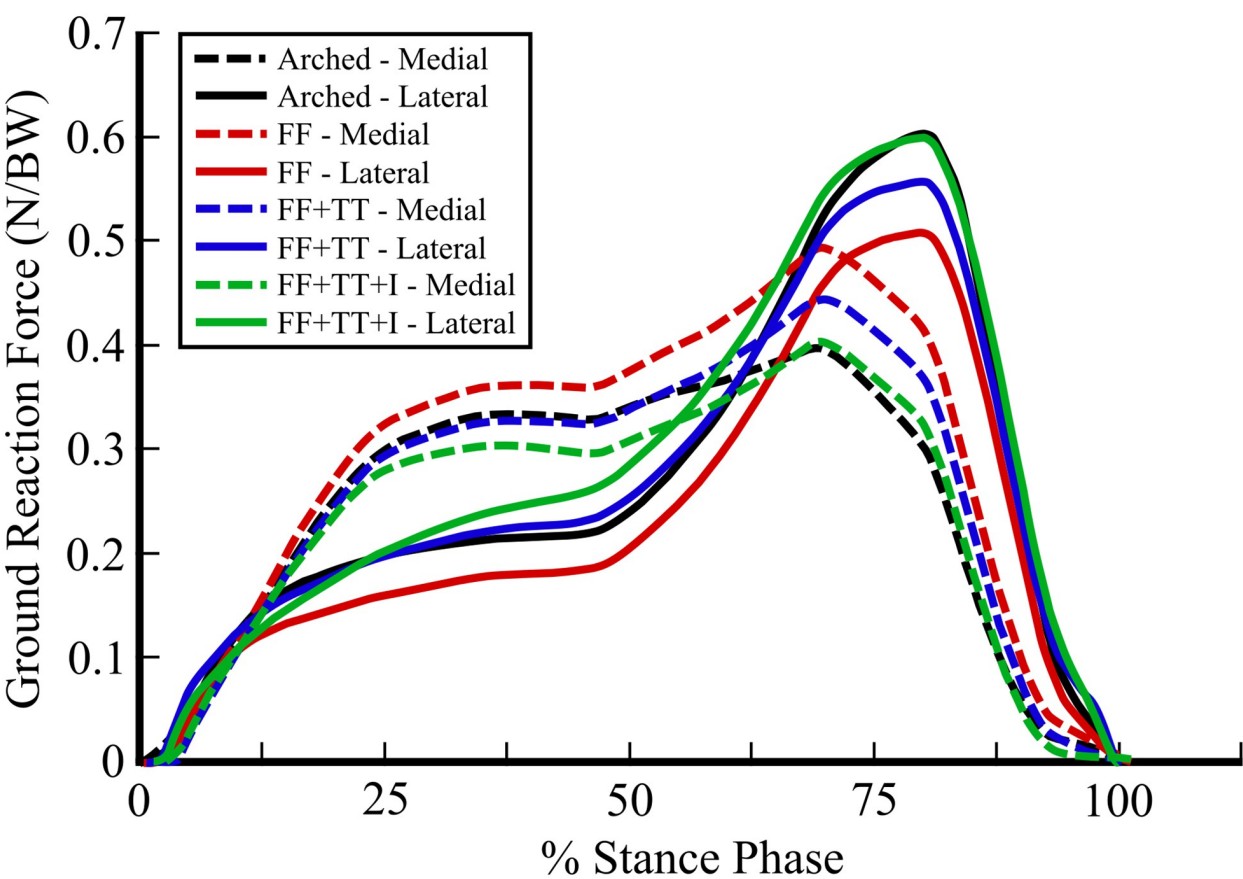

**Fig 10. Medial-lateral split of vertical ground reaction forces.** The medial/lateral split of vertical ground reaction forces generated by gait simulations using an arched foot (arched; black), flatfoot (FF; red), flatfoot with tendon transfer (FF+TT; blue), and flatfoot with implant-modified tendon transfer (FF+TT+I; green) model. The results are normalized by body weight (BW).

Equivalently, these results represent a 45.3% restoration of the physiological medial/lateral distribution of vertical ground reaction forces generated by our arched foot model (39.7%/60.3% medial/lateral split).

In the gait simulation using the flatfoot model with implant-based tendon transfer surgery, a 40.2%/59.8% medial/lateral split in peak vertical ground reaction forces was recorded. This 8.1% lateral shift in the distribution of vertical ground reaction forces from the untreated flatfoot model represents a 94.2% restoration of the physiologic medial/lateral distribution of vertical ground reaction forces generated by our arched foot model. Compared to the traditional tendon transfer model, the implant-based model provided 2.1X the restoration of the physiological vertical ground reaction force distribution. In the previous human cadaver experiment with the Robotic Gait Simulator, the center of pressure at the same percent stance phase (75%) was shifted laterally by 5.3% in the implant-based tendon transfer condition when compared to the flatfoot condition [11]. The increased restoration of the medial/lateral distribution of vertical ground reaction forces observed in our simulations compared to the experimental results can be attributed to the lack of soft tissue interactions and tendon friction in our idealized foot models.

## Discussion

In this study, we created simplified biomechanical models in OpenSim of an arched foot, a flatfoot, a flatfoot with tendon transfer surgery, and a flatfoot with implant-modified tendon transfer surgery. Using a fixed-tibia and moving ground-plane simulation methodology, we evaluated the vertical ground reaction forces produced by each model on five contact points of the foot. The arched foot and flatfoot models have demonstrated the capability of reproducing the characteristic double-peak profile of vertical ground reaction forces during the stance phase of gait and each exhibit an appropriate medial/lateral distribution of these forces. The implant-modified tendon transfer model produced an 8.1% lateral shift in peak vertical ground reaction forces, resulting in a 94.2% restoration of the physiological medial/lateral distribution of vertical ground reaction forces observed in the arched foot model. This represents a 2.1X greater lateral shift of vertical ground reaction forces when compared to the traditional tendon transfer model (3.9% lateral shift).

Adult acquired flatfoot deformity currently requires both a tendon transfer surgery and osteotomy to be fully corrective, with the primary purpose of correcting anatomical deformity and returning supporting forces to the foot arch, thereby restoring a physiological distribution of vertical ground reaction forces [1, 5–7, 11]. The long-term objective of this study is to develop a force-amplifying implant that provides additional surgical treatment options for patients with stage II adult acquired flatfoot deformity while maintaining or improving upon current patient outcomes. The 8.1% lateral shift of vertical ground reaction forces predicted by our implant-modified flatfoot model is promising when compared with computational models developed by Spratley et al. of surgical treatment for adult acquired flatfoot deformity [59]. Using complex rigid body models that incorporate soft tissue interactions, FDL tendon transfer surgery with medializing calcaneal osteotomy was simulated in flatfoot models, which found that the combined treatment provided a 9.0% increase in lateral forefoot loading force when compared to the untreated flatfoot model. Based on these results, the pulley-like implant used in this work demonstrates potential for amplifying the force produced by the transferred FDL and significantly shifting the distribution of vertical ground reaction forces more laterally when compared to current FDL tendon transfer surgeries without requiring an additional bony procedure. Thus, if a pulley-like implant could be used to create a FDL tendon network to better replicate physiological PT tendon forces, an additional osteotomy procedure may not be required to appropriately restore the foot arch in all cases of adult acquired flatfoot deformity requiring surgical intervention. A simpler surgery that does not require an additional bony procedure may help to provide additional surgical treatment options for adult acquired flatfoot deformity with stage II PT tendon dysfunction where a standard has yet to be established [5, 6, 61, 62].

The models developed in this work are differentiated from previous gait models because they preserve enough biomechanical and anatomical complexity to study how surgical treatments and varying pathology affect the distribution of vertical ground reaction forces during the stance phase of gait. While other models of human gait have been developed, they typically use ground reaction forces as an input to inverse dynamic simulation rather than generate it as an output. Many models are either too complex to dynamically simulate gait or too simple to enable anatomical modifications. More complex models that incorporate soft tissue interactions and passive ligaments of the foot are typically simulated with a static load rather than in dynamic gait due to the computational and time expense of doing so [23–25, 59]. Conversely, simple models can accurately reproduce ground reaction forces are often abstracted to the point that they lose anatomical definition [26–30].

The same simulation methodology presented here for adult acquired flatfoot deformity could also be modified and applied to other models of foot pathology, such as clubfoot, plantar fasciitis, and planovalgus [12, 55, 63–65], to evaluate treatment effectiveness through the ground reaction forces produced during gait. When compared with cadaveric or clinical studies, the time and effort efficiencies of biomechanical simulation provide an attractive alternative for initially evaluating the biomechanics of novel treatment options.

## Limitations

While computational simulations can provide valuable information for evaluating the biomechanical feasibility of surgical treatment options, they also have inherent limitations. Specifically in biomechanics-focused simulations, certain critical biological processes cannot be modeled and accounted for. For example, we are not able to evaluate how scar tissue formation and tendon abrasion affect the long-term success of a translating pulley-like implant in tendon transfer surgery. Furthermore, the compensatory adaptations of muscles, tendons, and other tissues to adult acquired flatfoot deformity are not easily analyzed in simulation and were not included in this study. The differences in muscle force generation and excursion may ultimately have substantial impact on the efficacy of our implant-based tendon transfer procedure. To properly address these concerns, human cadaver studies of soft tissue interaction and live animal implantation studies of healing effects, rehabilitation, and implant biocompatibility must be conducted.

Biomechanical simulations are also limited by computational complexity. Simplifying assumptions and model modifications must be made to run time-efficient simulations. However, the trade-off is that portions of complex human anatomy are oversimplified in the models, which limits the scope to which simulation results can be extrapolated to clinical settings. In our simplified foot models, implant interaction with soft tissue was completely neglected. Thus, while we may be able to describe a theoretical maximum force-amplification efficacy for our proposed implant, the definitive functional improvement provided by the implant in a clinical setting is still unclear. The contact between our foot models and the ground plane was also greatly simplified to enable reasonable simulation times and model complexity. Our foot models only have five distinct spheres of contact with the ground plane when in actuality the foot is in continuous contact with the ground during gait with infinite points of contact. Furthermore, the contact spheres retained a constant size and location between all four foot models even though medio-lateral plantar pressure distributions are expected to change before and after surgical intervention and due to anatomical variations [16, 56]. We also use a simplified arch joint in our foot models that removes significant complexity from the ligaments, muscles, and bones that form the actual medial longitudinal arch. These simplifications result in low-resolution foot-ground pressure data and potentially unrealistic arch interactions that prevent immediate claims of surgical treatment effectiveness. However, when solely comparing between the four models developed in this study that all use the same methodology, we can still make reasonable estimations on trends of medial/lateral vertical ground reaction force shifts between the different foot conditions and surgical treatment options.

Other aspects of the proposed implant-based surgery can be examined in biomechanical simulation but were not explored in this work. Since we simulated the transferred FDL muscle path using the original routing of the PT, we assumed perfect tendon routing in tendon transfer surgery, which in reality has varying effects on tendon tensioning, tendon routing, and the active portion of the muscle force-length curve. We also used low resolution muscle force data, which makes it more difficult to accurately evaluate the continuous impact of surgical treatment on shifting the distribution of vertical ground reaction forces between the provided data

points. Finally, we only evaluated a single, optimal configuration of the implant-based procedure. In reality, surgical decisions for the proposed implant-based procedure, such as the FDL tendon anchoring location and anchoring angle, must be made to optimize the effectiveness of the proposed treatment. These additional parameters can be explored in more detail in future biomechanical simulations.

## Conclusions

The implant-based tendon transfer model and simulation developed in this work have demonstrated the biomechanical feasibility of a pulley-like implant to amplify the force produced by the the FDL in tendon transfer surgery for treatment of stage II adult acquired flatfoot deformity. If successfully developed, such an implant could potentially be applied to many other tendon transfer surgeries where muscle weakness is an issue, such as spinal cord injury, volumetric muscle loss, and peripheral nerve injury [9, 10, 66–70]. This implant has the potential to not only improve post-surgical outcomes for patients with muscle weakness but also to increase the patient population eligible for tendon transfer surgery. Since a grade of muscle strength is typically lost in tendon transfer procedures [71], donor muscles must meet minimum strength grade requirements to be considered for transfer. By effectively amplifying the muscle strength grade of the donor muscle, the implant could be indicated for patients with weaker candidate donor muscles to qualify for tendon transfer surgery.

While these preliminary findings and potential applications are encouraging, there are many aspects of the proposed surgical procedure that need to be explored further to determine clinical feasibility of such a device. The implant must undergo additional evaluation both in biomechanical simulation and experimentally. Computational simulation can be utilzied to examine the unexplored biomechanical effects of suboptimal tendon routing, higher resolution muscle activation data, and variability in surgical technique. Studies in human cadavers and live animals will also be required to fully explore the effects of scar tissue formation, soft tissue interaction, and tendon abrasion on efficacy of the implant.

## Supporting information

**S1 Fig.**
(JPG)

## Author Contributions

**Conceptualization:** Hantao Ling, Ravi Balasubramanian.

**Data curation:** Hantao Ling.

**Formal analysis:** Hantao Ling.

**Investigation:** Hantao Ling, Ravi Balasubramanian.

**Methodology:** Hantao Ling.

**Project administration:** Ravi Balasubramanian.

**Supervision:** Ravi Balasubramanian.

**Validation:** Hantao Ling.

**Visualization:** Hantao Ling.

**Writing – original draft:** Hantao Ling.

**Writing – review & editing:** Hantao Ling, Ravi Balasubramanian.

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
