## [Decision Letter · Decision Letter 0]

11 Jan 2021

PONE-D-20-33947

A novel implantable mechanism-based tendon transfer surgery for adult acquired flatfoot deformity: validation using biomechanical simulation

PLOS ONE

Dear Dr. Ling,

Thank you for submitting your manuscript to PLOS ONE. After careful consideration, we feel that it has merit but does not fully meet PLOS ONE’s publication criteria as it currently stands. Therefore, we invite you to submit a revised version of the manuscript that addresses the points raised during the review process.

The reviewers have asked several methodological questions including model detail and validity.  Additionally, what are the clinical contraindications to this approach?  The PTT is often of significantly compromised quality, not the muscle, as others have reported including Dr. Deland.  This means that the tissue is incapable of sustained loading and would likely fail with the device you are designing. Further does scarring around the implant and/or tendon prevent tendon excursion to fully realize the mechanical advantage potentially provided by the device?

We look forward to receiving your revised manuscript.

Kind regards,

Jennifer S. Wayne

Academic Editor

PLOS ONE

Journal Requirements:

2. We note that Figure 1 in your submission contains copyrighted images. All PLOS content is published under the Creative Commons Attribution License (CC BY 4.0), which means that the manuscript, images, and Supporting Information files will be freely available online, and any third party is permitted to access, download, copy, distribute, and use these materials in any way, even commercially, with proper attribution. For more information, see our copyright guidelines: http://journals.plos.org/plosone/s/licenses-and-copyright.

(1) You may seek permission from the original copyright holder of Figure 1 to publish the content specifically under the CC BY 4.0 license.

3. We noted in your submission details that a portion of your manuscript may have been presented or published elsewhere. ("Portions of the models and results of this paper have been presented in short abstracts at the American Society of Biomechanics conferences in 2016 and 2017. ") Please clarify whether this conference proceeding was peer-reviewed and formally published. If this work was previously peer-reviewed and published, in the cover letter please provide the reason that this work does not constitute dual publication and should be included in the current manuscript.

Reviewers' comments:

Reviewer's Responses to Questions

**Comments to the Author**

1. Is the manuscript technically sound, and do the data support the conclusions?

Reviewer #1: Partly

Reviewer #2: Partly

2. Has the statistical analysis been performed appropriately and rigorously? 

Reviewer #1: N/A

Reviewer #2: N/A

3. Have the authors made all data underlying the findings in their manuscript fully available?

Reviewer #1: Yes

Reviewer #2: Yes

4. Is the manuscript presented in an intelligible fashion and written in standard English?

Reviewer #1: Yes

Reviewer #2: Yes

5. Review Comments to the Author

Reviewer #1: This study developed musculoskeletal models of the foot and ankle with the overall aim of assessing a newly proposed surgical intervention. My main concerns about the manuscript relate to the modeling process, with much more detail required here.

Lines 7-9: Assuming you are using Johnson and Strom classification, this should be cited. Also, it should be clarified that at stage 2 the deformity is still flexible enough to be corrected

Line 19: My understanding is that the osteotomy may also be required depending on how much flexibility there is in the deformity, in some cases it may not be a matter of being able to generate enough force

Line 22: This paragraph could be clearer about what stage of development the proposed implant is at, from the following paragraph and the discussion it appears that it’s only been tested in a cadaver arm?

Line 33-35: Note here that the pulley also reduces the range of motion of the muscle, it's not simply doubling the force for free

Line 61: I'm not familiar with the term "foot's frontal axis"; I assume this refers to the sagittal plane of the foot?

Lines 138-141: Not clear what the degrees of freedom of the new joints are and how they interact with the other segments? More detail required here.

Line 149: What properties were used for the deltoid and spring ligaments?

Lines 184-186: Limitation is that there are likely compensatory changes in the activity of other muscles in FF conditions, these are not accounted for

Line 200: Did you try non-linear interpolation for these? The true activation patterns are unlikely to be linear between points

Line 214: The RGS is limited to this speed by physical constraints, did you consider increasing the speed of your simulation back to normal?

Line 221: Most simulations of the foot now use more complex models that can account for the non-linear viscoelastic behavior of the foot tissue.

Line 225: Deformations are likely very different between sites on the foot

Line 225-226: More details on this needed about this process. Was this a static simulation? This one of the key parts for tuning the models and how it was performed makes the validation data more or less externally convincing

Line 229: Vs symbol different in the equation and definitions? And what value was used for Fv?

Line 247-248: How did the AP and ML reaction forces look?

Lines 280-290: I’m not sure how directly comparable measurements of plantar pressure are to the discrete contact spheres used in these models

Reviewer #2: PONE-D-20-33947 - Review

Ling, Hantao; Balasubramanian, Ravi

“A novel implantable mechanism-based tendon transfer surgery for adult acquired flatfoot deformity: validation using biomechanical simulation"

Summary:

The authors present an adapted rigid body model (OpenSim) of the human lower extremity in states representative of normal, flatfoot, and two reconstructed flatfoot states. They further elaborate on how the medial/lateral load distribution is changed between the model states under conditions simulating dynamic single-leg stance gait.

The manuscript addresses a question with scientific merit and is well written. However, much detail is necessary to understand how the model was constituted and what assumptions were made with regard to the mechanics simulated. Finally, the study is more appropriately presented as an exploration of a potential new device. It is not clear what if any quantification is made of model biofidelity relative to the in vivo case.

General Comments:

1. Model simplicity

a. The OpenSim model is very simple and while segment geometry is visualized, it has no effect on joint function. This does not diminish potential model utility, but the predictive quality of the model should be contextualized relative to these simplifications

b. The referenced OpenSim model is built to mimic normal lower extremity normal joint function. The appropriateness of using ‘normal’ joint and muscle definitions when simulating gross deformity should be included and contextualized

i. STRUCTURE

1. The model is modified to mimic a flat foot arch by manually displacing the medial column segment of the model. This displacement is maintained through a torsional spring. It is not clear how this affects foot mechanics and no comparison to in vivo response is given.

2. More detail would help understanding how the path of the transferred ‘FDL’ was achieved. It seems that by using the existing PTT model elements and simply assigning it the force trajectory of the FDL, you have neglected any effects of routing. Is this correct? Why is this justified?

3. Needs figure of model

a. Why constrained to sagittal?

ii. TENDON

1. the muscle curves are directly applied as force, which assumes that not only is the activation unchanged but that the structural deformity (with accompanying tendon rerouting & length changes) has no effect on the force generated at a given % of the stance phase. Is it reasonable / useful to use these ‘normal’ inputs for this deformed case? Please justify.

iii. MUSCLE

1. The in vivo muscles have finite contraction lengths which are proportional to the number of sarcomeres in series in the muscle belly. While these can remodel to accommodate within limited ranges to effect new contractile lengths, the overall contraction length of the FDL is relatively constant. The pully mechanism incorporated does indeed have the effect of doubling force output… but it does so at the expense of halving the excursion of the tendon.

2. It is not clear that this limitation is reflected in the model

3. It is not clear that the full dorsiflexion � plantar flexion ROM would be achievable if the effective excursion of the tendon were halved

2. Much more detail is needed on the model. While some details can be obtained through the provided OpenSim references, much more is needed to explain HOW the model is used:

a. For example, the boundary conditions assigned to the model and which parameters were driven, which were reported, which if any were used in the control of the platform (this is done in the referenced robotic cadaver work… was it similarly implemented in the model?), which joint degrees of freedom were allowed and at what locations, what are the stiffnesses and permitted ranges of those joints, etc.

b. How are the inertial properties of the limb implemented and at what rate are the motions simulated?

i. In particular, it is not clear how the axial force of the model was implemented during the simulation. The initial position seemed to be tuned by incrementally lowering the vertical plantar surface of the foot toward loading platform until the reaction force generated by the sphere/platform penetration was approximately equal to BW. After initial positioning though it is not clear what was done.

1. Since it is referenced as a forward dynamics simulation, presumably there is a dynamic axial force and the vertical displacement is free to change in order to maintain equilibrium. Is this correct?

ii. The characteristic bimodal shape of the GRF curve captured for living subjects is the result of deceleration of their body mass at heel strike and acceleration during toe off. Here the subject body mass doesn’t change in time during stance phase and rate of loading is critical. Due to the significant technical challenges with control and instability many (all?) forward dynamics experimental and model simulations that I’m aware of are typically executed much slower than what is reflected in the GRF curves in Fig 8.

c. The above comments (a,b) are relevant because the GRF is reported as an output and used as demonstration of the models’ fidelity. However, it seems that this is perhaps a controlled input. *IF* that is the case, its agreement is simply verification of the inputs and should not be used as part of the objective scoring of the model.

3. Validation versus exploration

a. The model provides useful insight into flatfoot function and the mechanics of how tendon transfer reconstructions may work

b. However, it is not clear what reference human or cadaver data is presented and how the model biofidelity is scored relative to it.

i. See above comments with regard to the use of GRF

ii. Figure 9 shows the sensitivity of the model to incorporation of the TT and TT+I. However, the reference normal (Black & Black Dashed) lines are also from the simulation. It is not clear how claims of biofidelity are judged here since all comparisons are between states of the same model under review.

1. The magnitude of changes (medial load shifted to lateral) should be discussed relative to those reported in the literature for human or cadaveric experiments. References should be expanded to include such prior work.

Specific Comments:

181-183 Exploring *all* the sensitivities of the model is perhaps out of scope. However, to say that *any* exploration of sensitivity is out of scope is not reasonable. The work may show the promise of the technique but claims that the model is ‘validated’ are overstated.

224-225 This is the first mention of the any soft-tissue element in the model. How was the footpad implemented? Were there any other soft tissues (e.g., skin, fat, muscle, etc.?)

If the full mass of soft tissues were not incorporated, how were the limbs inertial properties approximated?

237-238 Additional detail regarding how contact sphere diameter was derived from studies of foot shape

344-345 See General Comment 2. Detail should be added to Methods to elaborate on this point.

Figure Comments

FIG3 Suggest elaborating/expanding this figure with detail about the segments, joints, and contact spheres.

FIG4 This figure spot may be better used as a place to show the specifics of the device and resulting tendon routing in the various model states

FIG5 Label the subplots B,C to reflect they are input kinematics

FIG8 Caption Denote that the black line is experimental data(?) (cadaver or human subject)

FIG# A simulation matrix would help in showing what was simulated and what model attributes were changed to represent a given state

6. PLOS authors have the option to publish the peer review history of their article (what does this mean?). If published, this will include your full peer review and any attached files.

Reviewer #1: No

Reviewer #2: No

---

## [Author Response · Author response to Decision Letter 0]

3 Aug 2021

The Response to Reviewers is attached as a PDF in the submission.

---

## [Editor Report · Decision Letter 1]

3 Aug 2021

PONE-D-20-33947R1

A novel implantable mechanism-based tendon transfer surgery for adult acquired flatfoot deformity: evaluating feasibility in biomechanical simulation

Dear Dr. Ling,

The above-mentioned manuscript has now been withdrawn from the review process at PLOS ONE.

If this was a mistake, please do reach out to us at plosone@plos.org. Otherwise, no further action is needed from you at this point.

Sincerely,

PLOS ONE

---

## [Author Response · Author response to Decision Letter 1]

13 Sep 2021

The response to the reviewers is attached in the 'Response to Reviewers.pdf' file included in this submission.

---

## [Decision Letter · Decision Letter 2]

25 May 2022

PONE-D-20-33947R2

A novel implantable mechanism-based tendon transfer surgery for adult acquired flatfoot deformity: evaluating feasibility in biomechanical simulation

PLOS ONE

Dear Dr. Ling,

Thank you for submitting your manuscript to PLOS ONE. After careful consideration, we feel that it has merit but does not fully meet PLOS ONE’s publication criteria as it currently stands. Therefore, we invite you to submit a revised version of the manuscript that addresses the points raised during the review process.

We look forward to receiving your revised manuscript.

Kind regards,

Amarjit Singh Virdi, PhD

Academic Editor

PLOS ONE

Journal Requirements:

Reviewers' comments:

Reviewer's Responses to Questions

**Comments to the Author**

1. If the authors have adequately addressed your comments raised in a previous round of review and you feel that this manuscript is now acceptable for publication, you may indicate that here to bypass the “Comments to the Author” section, enter your conflict of interest statement in the “Confidential to Editor” section, and submit your "Accept" recommendation.

Reviewer #1: All comments have been addressed

Reviewer #3: (No Response)

2. Is the manuscript technically sound, and do the data support the conclusions?

Reviewer #1: Yes

Reviewer #3: Partly

3. Has the statistical analysis been performed appropriately and rigorously? 

Reviewer #1: Yes

Reviewer #3: N/A

4. Have the authors made all data underlying the findings in their manuscript fully available?

Reviewer #1: Yes

Reviewer #3: Yes

5. Is the manuscript presented in an intelligible fashion and written in standard English?

Reviewer #1: Yes

Reviewer #3: Yes

6. Review Comments to the Author

Reviewer #1: The authors have addressed all of my comments in this much improved manuscript. Congratulations on an interesting study.

Reviewer #3: The authors have addressed most of the comments in the previous round of review. Several minor revisions are necessary.

Related to comments by previous reviewers:

Both reviewers made comments that the intervention potentially reduces the range-of-motion (ROM) of the muscle (Reviewer 1) and the ankle joint (Reviewer 2). The authors revised and explained well why such an excursion trade-off is necessary to generate force that is 2X than what the FDL muscle is capable of. They also clarified that the full ROM from dorsiflexion to plantarflexion was simulated. The simulated ROM was passive (produced by the moving ground plate), while the previous reviewers’ comments seem to concern the active ROM. The authors are recommended to discuss the potential effect or limitation on ROM to directly address the reviewers’ comments.

Reviewer 2 suggested to clarify inputs and outputs of the simulation and what reference human or cadaver data is presented. The authors revised by including additional references. Citation 35, 36, and 37 show that the RGS has been used to simulate physiological and pathological gait. By reviewing citation 11, the kinematics of the moving ground plate seems to come from the experimental data (output), which was generated by that tibia to ground kinematics of healthy participants and experimentally induced flat foot models (inputs). I understand it is difficult to classify whether the ground plate kinematics is physiological or pathological. What about the EMG estimates (line 252-254)? Do they come from healthy participants or not (line 252-254; citation 35, 42, 43)? If the data come from pathological population, what exactly are the pathologies?

Additional comments by current reviewer:

The same dimension of contact spheres were used for all four foot models. However, as suggested by citation 16 and 56, medio-lateral plantar pressure distribution can change before and after surgical intervention and between individuals presenting anatomical variations. This limitation is suggested to be discussed like the other model simplicity limitations (e.g., compensatory muscle activations).

The medial and lateral vertical ground reaction force in lines 385-395 do not completely match what are shown on figure 10. For example, in text, the arched foot showed medial and lateral 0.454 and 0.691 N/BW, respectively. However, on the figure, the respective medial and lateral peaks are about 0.4 and 0.6.

Line 46-50 regarding excursion trade-off may be moved and combined with lines 233-237.

Line 40-42 “…amplify the force transferred from the weaker FDL to the PT [tendon]”

Figure 8 in text shows gait events sequentially from left to right. It would be an improvement if a left lower leg is shown as compared to a right leg, allowing readers to visualize the foot proceeding the stance phase of gait from left to right. However, I understand that the authors might intend to show the medial side of the foot.

In summary, the revision includes additional information, addressing most of the suggestions by the previous reviewers. With the suggested minor revisions, the manuscript would be a sound piece of scientific research and recommended to be accepted by the journal.

7. PLOS authors have the option to publish the peer review history of their article (what does this mean?). If published, this will include your full peer review and any attached files.

Reviewer #1: No

Reviewer #3: No

---

## [Author Response · Author response to Decision Letter 2]

8 Jun 2022

The reviewer response is provided in the 'Response to Reviewers.pdf' document.

---

## [Editor Report · Decision Letter 3]

15 Jun 2022

A novel implantable mechanism-based tendon transfer surgery for adult acquired flatfoot deformity: evaluating feasibility in biomechanical simulation

PONE-D-20-33947R3

Dear Dr. Ling,

We’re pleased to inform you that your manuscript has been judged scientifically suitable for publication and will be formally accepted for publication once it meets all outstanding technical requirements.

Kind regards,

Amarjit Singh Virdi, PhD

Academic Editor

PLOS ONE

Additional Editor Comments (optional):

Thank you for your prompt response to the reviewer's comments which improved (and corrected) the manuscript for the readers.
---

## [Editor Report · Acceptance letter]

17 Jun 2022

PONE-D-20-33947R3 

A novel implantable mechanism-based tendon transfer surgery for adult acquired flatfoot deformity: evaluating feasibility in biomechanical simulation 

Dear Dr. Ling:

I'm pleased to inform you that your manuscript has been deemed suitable for publication in PLOS ONE. Congratulations! Your manuscript is now with our production department. 

Kind regards, 

on behalf of

Dr. Amarjit Singh Virdi 

Academic Editor

PLOS ONE